# Physical Activity through a Classroom-Based Intervention: A Pragmatic Non-Randomized Trial among Swedish Adolescents in an Upper Secondary School

**DOI:** 10.3390/ijerph182111041

**Published:** 2021-10-20

**Authors:** Filip Christiansen, Viktor H. Ahlqvist, Mikaela Nyroos, Hans Löfgren, Daniel Berglind

**Affiliations:** 1Department of Global Public Health, Karolinska Institutet, 171 77 Stockholm, Sweden; viktor.ahlqvist@ki.se (V.H.A.); daniel.berglind@ki.se (D.B.); 2Department of Education, Umeå University, 901 87 Umeå, Sweden; mikaela.nyroos@umu.se; 3Department of the Police Education Unit, Umeå University, 901 87 Umeå, Sweden; hans.lofgren@umu.se; 4Centre for Epidemiology and Community Medicine, Region Stockholm, 104 31 Stockholm, Sweden

**Keywords:** physical activity, school-based intervention, accelerometer

## Abstract

Schools are an important arena to curb the decline in physical activity (PA) in youth. School-based interventions with accelerometer-measured PA are warranted. This study aimed to increase accelerometer-measured PA in adolescents following a 12-month school-based intervention. Two school-classes of 16–18-year-old Swedish students were allocated to intervention group and control group. Accelerometer-measured PA was gathered at baseline, 6- and 12-month follow-up. Mixed-effects linear regression was used to investigate between-group and within-group differences in mean minutes per day (min/day) of moderate to vigorous PA (MVPA), light PA (LPA) and sedentary time (ST). Fifty-seven students participated (intervention group = 31, control group = 26). At 12-month follow-up, the intervention group performed 5.9 (95% CI: −4.3, 16.2) min/day more in MVPA, 1.8 (95% CI: −17.9, 14.2) min/day less in LPA, and 4.1 (95% CI: −27.3, 19.2) min/day less in ST compared to the control group. Within the intervention group, there was no significant change in PA. Within the control group, LPA decreased (95% CI: −19.6, −0.2; *p* = 0.044) and ST increased (95% CI: 1.8, 30.8; *p* = 0.028). Although no between-group differences in PA were statistically significant, the within-group changes may suggest a preventive impact on the decline in PA during adolescence.

## 1. Background

Physical activity (PA) has a beneficial effect on mental and physical health [1], whereas excessive sedentary time (ST) is a risk factor for morbidity in adults [2]. The transition from childhood to adolescence is characterized by increasing ST and a decrease in PA [3]. Notably, both globally and in Sweden, a majority of adolescents fail to achieve the World Health Organization (WHO) recommendation of at least 60 min of moderate to vigorous PA (MVPA) per day [4,5]. Since PA in childhood and adolescence predicts PA in adulthood [6,7], and since PA of any intensity is associated with a substantial reduction of all-cause mortality in a dose-response pattern in adults [8], finding ways to increase PA in youth could have a positive long-term impact on public health [9]. 

Adolescents spend a large proportion of their waking hours at school [10], where ST is predominant [11]. Since schools can reach a majority of adolescents, the school environment is an important arena to promote PA in youth [12]. However, previous school-based interventions to increase PA in adolescents have shown modest positive results. For example, a meta-analysis by Borde et al, including 12 randomized controlled school-based PA interventions in adolescents, reported a standardized mean difference (SMD) of 0.24 min in MVPA between intervention and control groups (95% CI: −0.08, 0.56) [13]. Similarly, Love et al. included 17 cluster-randomized controlled school-based PA interventions and found a SMD of 0.02 min in MVPA in between-group comparisons (95% CI: −0.07, 0.11) [12].

Another problem is the frequent use of self-report methods to measure PA in previous studies, which are known to overestimate PA while underestimating ST compared to objective measures [14]. For example, in a systematic review by Hynynen et al., four out of seven included randomized school-based interventions that reported significantly positive effects on PA were solely based on self-report data [15]. An additional challenge is that relatively few interventions have targeted older adolescents [13,15].

Considering the importance of preventing a decline in PA in late adolescence, and the scarce amount of previous school-based interventions with objectively measured PA in this age group, we report the findings of a pragmatic trial that aimed to increase accelerometer-measured PA in a group of students aged 16–18 years. Specifically, the trial was set in a Swedish upper secondary school during 2019 and aimed to increase accelerometer-measured PA at 12 months following an experimental school-based intervention, with the long-term ambition to also promote academic performance and mental health. 

## 2. Methods

### 2.1. Study Design and Participants

The “Active and Participating by Innovative Classrooms” study was a pragmatic school-based intervention study set in an upper secondary school in a small city in the middle of Sweden during 2019. In January 2019, a group of 16–17-year-old (age at recruitment) students attending the first year of the same upper secondary program and grade but in two separate school classes, were invited to participate. Parental consent was obtained in a separate information meeting in January 2019. The school-classes were pre-determined, by the local project coordination group, to be an intervention group and a control group respectively, resulting in two non-randomized groups. Specifically, in the selection procedure of intervention vs control group condition, it was noted that one of the two school classes consisted of students attending additional physical education (PE) (2–3 additional PE-classes per week). These additional classes of PE were part of an extended PE-program at the school, and thus, were similar to the ordinary PE-classes in terms of content and duration. The school class with a higher frequency of PE-classes was assigned the control condition, emulating an active control group with expected greater baseline PA, and maximizing potential health benefits for the least physically active class. The trial was pre-registered [16], and approved by the Swedish Ethical Review Authority (Reference Number: 2019-01128).

### 2.2. Instrumentation and Procedure

The intervention group was exposed to a PA-promoting intervention consisting of (1) activity breaks during lectures in English and Mathematics, (2) adjustments to the classroom environment, and (3) time for reflection during PE classes (Figure 1). The multi-component intervention approach was based on theories of social-ecological influence on behavioral patterns, linking individual change to the social and physical environment [17], which have been shown to be effective to increase PA in youth [18]. The total length of exposure was 14 weeks during the spring semester and 14 weeks during the fall semester of 2019. 

During lectures in English and math, the exposure consisted of (1) low-intensity PA breaks of 5 min twice per lecture, and (2) classroom adjustments consisting of standing tables, desk bikes, and additional equipment such as balance pads, exercise balls, and elastic resistance bands. Activity breaks served to interrupt longer sessions of ST, during which the additional equipment served to facilitate unstructured PA, and the use of desk bikes can increase levels of LPA [19]. Lectures in math were scheduled three times per week, and in English two times per week, all extended by 10 min to enable activity breaks (i.e., a total extension of 50 min per week). To ensure an equal time distribution between standing tables, desk bikes, and regular sitting tables respectively, the intervention group was divided into three groups and assigned an equipment type for two weeks followed by rotations.

Finally, during PE classes once per week, the intervention consisted of (3) time for reflection on the intervention and the impact of PA on physical and mental health in general, to increase commitment and participation [20]. To facilitate these reflections, the intervention group listened to 15-min podcasts with inspirational content while walking outdoors during PE classes once every two weeks.

### 2.3. Measures

The trial involved three measurement points with accelerometer-measured PA; baseline assessment in January, mid-intervention (6 months) in June, and 12-month follow-up in December. PA was measured using Actigraph GT3X accelerometers, providing greater reliability and validity compared to subjective assessments [21]. Recommended settings for wear protocol were considered; measures were made with accelerometers at the right hip during waking hours for seven consecutive days at all measurement points [22]. Minimum wear time was set to ≥10 h per day and non-wear time was defined as ≥60 min with zero activity counts (i.e., no accelerations due to body movement) [22]. A valid measurement period was defined as ≥1 day with ≥ 10 wear time hours. Activity counts were analyzed as vector magnitude (Vm) in 15-s epochs, in line with current recommendations for children and adolescents [22]. PA-intensities were calculated based on recommended cut-off values [23]: ST was defined as <720 counts per minute (CPM), LPA as 721–3027 CPM, MPA as 3028–4447 CPM, and VPA as ≥4448 CPM. Raw accelerometer data files were processed using ActiLife software version 6.13.4.

Descriptive characteristics were collected from questionnaires at all measurement points, where participants were asked to detail their date of birth, sex (female/male), organized sports participation (yes/no), distance to school (kilometers), season-dependent active commuting (bicycling/walking all year, during spring or fall semester, or not at all), and describe if they had a foreign background (born outside Sweden or both parents born outside Sweden). 

### 2.4. Data Analysis

Descriptive characteristics and baseline accelerometer data on PA are presented using relevant measures of central tendency and dispersion. All analyses were performed using an intention-to-treat analysis. The primary outcome was between-group differences in MVPA (mean minutes per day) at 6- and 12-month follow-up. Secondary outcome measures included between-group differences in LPA and ST (mean minutes per day), and within-group changes of each activity intensity. To model the mean activity in each group at each time point and its change over time we employed linear mixed-effects regression. To account for the repeated nature of data (i.e., the same participants over time), all models were fit with a random intercept for individuals and with robust standard errors. Each intensity of PA was analyzed in separate models and all models were adjusted for sex and accelerometer wear time hours. All statistical tests were two-sided and a *p*-value < 0.05 was considered statistically significant. All data analyses were performed in Stata version 15.1 (StataCorp, College Station, TX, USA).

## 3. Results

### 3.1. Participants and Baseline Characteristics

Of the two enrolled classes, a total of 61 students chose to participate of which 32 were in the intervention group and 29 in the control group. At baseline measurement, valid accelerometer data were provided by 31 participants in the intervention group and 26 participants in the control group (Table 1). At 12-month follow-up, valid accelerometer data were provided by 26 and 25 participants in the intervention and control group, respectively.

The intervention group had a high proportion of females (80.6%) compared to the control group (57.7%), and the control group spent a slightly higher proportion of accelerometer wear time in MVPA and LPA at baseline. Baseline accelerometer wear time was similar in both groups at 13.3 (SD 1.5) and 14.0 (SD 1.1) hours/day in the intervention and control group, respectively. Mean accelerometer wear time remained similar in the groups at mid-intervention (6 months) (13.3 (SD 1.2) and 13.0 (SD 1.1) hours/day in the intervention and control group, respectively) and 12-month follow-up (12.8 (SD 1.6) and 13.3 (SD 1.2) hours/day in the intervention and control group, respectively).

### 3.2. Between-Group Differences in PA

At the baseline measure, the intervention group performed approximately 3.9 (95% CI: −13.7, 5.9; *p* = 0.437) minutes/day less MVPA compared to the control group (Figure 2). At 6- and 12-month follow-up, the intervention group performed 2.0 (95% CI: −9.5, 13.5; *p* = 0.732) minutes/day and 5.9 (95% CI: −4.3, 16.2; *p* = 0.255) minutes/day more MVPA compared to the control group, respectively. 

There was a statistically significant difference in both secondary outcomes at baseline (Figure 2); the intervention group performed less LPA (95% CI: −34.3, −2.8; *p* = 0.021) and more ST (95% CI: 0.7, 44.0; *p* = 0.043). However, there were no statistically significant between-group differences in mean minutes per day of ST and LPA at either 6-month or 12-month follow-up. Specifically, at 6 months the intervention group performed 12.6 min/day less LPA (95% CI: −29.8, 4.6; *p* = 0.152) and 10.4 min/day more ST (95% CI: −12.9, 33.7; *p* = 0.380) compared to the control group. At 12-month follow-up the intervention group performed 1.8 min/day less LPA (95% CI: −17.9, 14.2; *p* = 0.823) but, contrary to 6 months, 4.1 min/day less ST (95% CI: −27.3, 19.2; *p* = 0.732) compared to the control group.

### 3.3. Within-Group Changes in PA

In the intervention group, the mean minutes of MVPA increased across all measurement points, resulting in an estimated increase of 3.3 (95% CI: −3.9, 10.6; *p* = 0.367) minutes/day from baseline to 12-month follow-up (Table 2). Contrary, in the control group the mean minutes of MVPA decreased across all measurement points with an estimated decrease of 6.5 (95% CI: −14.3, 1.3; *p* = 0.102) minutes/day at 12 months.

From baseline to 12-month follow-up, the intervention group increased LPA by 6.8 (95% CI: −4.2, 17.8; *p* = 0.228) min/day and decreased ST by 10.1 (95% CI: −25.6, 5.4; *p* = 0.203) min/day, whereas the control group decreased LPA by 9.9 min/day (95% CI: −19.6, −0.2; *p* = 0.044) and increased ST by 16.3 min/day (95% CI: 1.8, 30.8; *p* = 0.028) (Table 2).

## 4. Discussion

We have presented the findings of a pragmatic trial that aimed to increase accelerometer-measured PA in a group of students aged 16–18 years following a 12-month experimental school-based non-randomized intervention. There were not any statistically significant between-group differences in MVPA (main outcome), LPA or ST (secondary outcomes), at any of the measurement points. However, from baseline to 12-month follow-up, the intervention group increased time spent in accelerometer-measured MVPA and LPA and decreased in ST, whereas the opposite was observed in the control group. 

The observed within-group changes may reflect the proposed transition towards increased ST, which was observed in the control group but not in the intervention group. In specific, the control group significantly reduced time spent in LPA by 9.9 min/day (95% CI: −19.6, −0.2; *p* = 0.044) and increased time in ST by 16.3 min/day (95% CI: 1.8, 30.8; *p* = 0.028). As such, our secondary findings suggest that the intervention may have had a preventive impact on the transition from PA to ST that typically occurs during adolescence [3]. 

Nevertheless, the positive but not statistically significant between-group effects on PA in the present study are in line with several previous school-based PA interventions [12,13]. Notably, a large proportion of previous school interventions reporting positive effects on PA are based on subjective measures (e.g., questionnaires) [12,24]. However, in contrast to subjective measures of PA, studies utilizing objective measures of PA have failed to demonstrate any effect of school-based interventions [12]. A potential reason could be the risk of overestimating PA with subjective measures compared to objective methods [14]. Therefore, comparisons with results from studies with subjective measures are challenging as the findings may differ due to measurement reliability and validity. An additional challenge is that relatively few previous interventions have targeted older adolescents specifically, since the effectiveness of interventions may vary between age groups [15].

## 5. Strengths and Limitations

A major strength of this pragmatic trial was the use of accelerometers to objectively measure PA, with greater reliability and validity as compared to self-report methods [22]. However, accelerometers have a limited ability to capture movements such as swimming, weight training, or bicycling, which may result in an underestimation of PA [25]. The use of accelerometers also implies attaching a foreign element to the body, which could cause compliance issues in the event of discomfort or a lack of knowledge on how to use the equipment. Hence, careful how-to-use instructions and appropriate preparation of the equipment is important to limit the risk of reduced accelerometer wear-time. An additional strength was a comprehensive multi-component intervention, including adjustments to the physical environment as well as curricular activities with reflections, walking podcasts, and increased recess time, since multicomponent intervention approaches are most effective in promoting PA in adolescents [26]. Furthermore, the 12-month duration of the trial included measures across seasonal variations, which increases the reliability of the findings as objectively measured MVPA in adolescents is higher during the spring [27].

However, the non-randomized small-scale nature of the pragmatic trial has several limitations. The study population consisting of 57 individuals was relatively small compared to previous school-based interventions, making analyses particularly vulnerable to invalid accelerometer data and/or insufficient wear time. To maximize the sample size, we decided to adjust for wear time rather than exclude participants with insufficient wear time according to traditional inclusion criteria, typically defined as ≥3 days with ≥10 wear time hours/day [21]. This is an acknowledged limitation, as the relaxed inclusion criteria increase the power at expense of the reliability of estimated time in different PA-intensities [22]. Furthermore, there was a substantial difference in sex distribution between the groups, particularly skewed in the intervention group consisting of 80.6% females. Since females engage in less PA than males during adolescence [28], this could contribute to the higher levels of PA and lower ST observed at baseline in the control group. Although all models were adjusted for sex, our findings are likely affected by residual confounding, and as such our findings should be interpreted with caution. 

In addition, some circumstances regarding the implementation and execution of the intervention are important to consider. The project leader reported that students tended to choose certain table types depending on individual preferences, implying that the 6-week intervention cycles with different equipment were not perfectly performed according to the protocol during the whole intervention. Although the use of a desk bike or standing position has limited representation in hip-worn accelerometer-measured PA, since the registration of such activities is weak [25], it might influence individual change in PA over time. This is due to the theoretical relation between behavioral change and the physical environment, which is a central idea of the present intervention approach [17].

## 6. Implications for School Health

Schools are important to prevent the decline in PA in adolescents and provides an opportunity to promote positive long-term health behaviors. In general, efforts to increase and measure PA objectively are expensive and resource-demanding, and few previous interventions have focused solely on late adolescents. Hence, this pragmatic study that aimed to increase accelerometer-measured PA in a group of students aged 16–18 years is a valuable contribution, adding to the pool of objective data on PA in this age group and describes certain challenges in changing the physical school environment to increase PA. Although the present intervention study did not result in a significant increase in PA, it highlights the importance of actions to promote PA as the control group may reflect the typical transition to more ST during adolescence. In addition, the results suggest that an experimental intervention approach in a small-scale setting may have a preventive impact on the decline in PA. This potential benefit should be further explored in future research using larger sample size and randomization

## 7. Conclusions

Following a 12-month experimental school-based intervention there were not any statistically significant between-group differences in MVPA, or LPA and ST. Yet, the within-group changes may suggest that the intervention had a preventive impact on the transition from PA to ST that typically occurs during adolescence.

## Figures and Tables

**Figure 1 ijerph-18-11041-f001:**
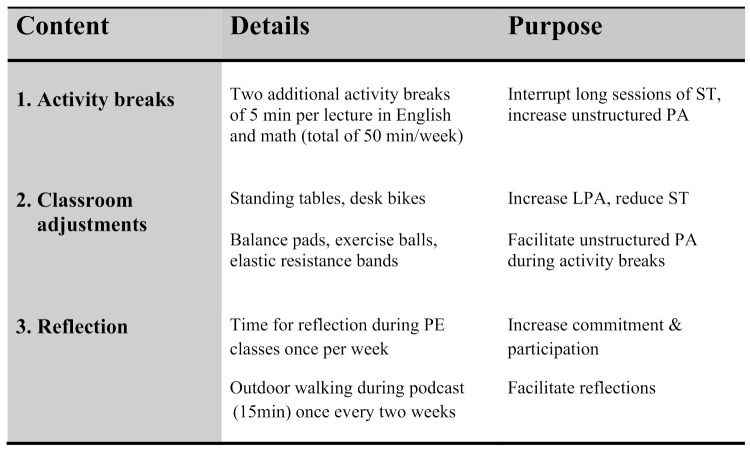
Overview of the Intervention Content. Abbreviations; PA = physical activity, LPA = light physical activity, ST = sedentary time, PE = physical education.

**Figure 2 ijerph-18-11041-f002:**
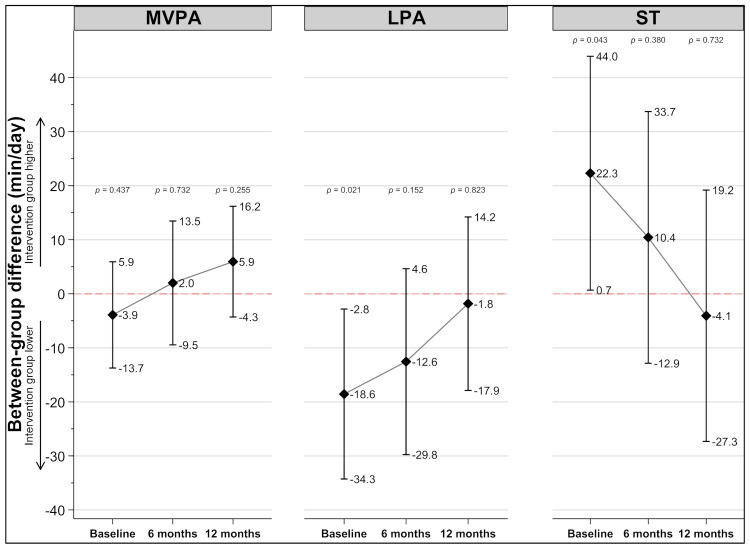
Between-group differences in physical activity and sedentary time from baseline to 12-month follow-up. Adjusted for sex and accelerometer wear time. Abbreviations; MVPA = moderate to vigorous physical activity; LPA = light physical activity; ST = sedentary time.

**Table 1 ijerph-18-11041-t001:** Characteristics of the study population and crude accelerometer data at baseline (T1).

	Total	Control	Intervention
	N = 57	N = 26	N = 31
Age, mean (SD)	16.1 (0.4)	16.1 (0.3)	16.2 (0.5)
Sex			
Female, n (%)	40 (70.2%)	15 (57.7%)	25 (80.6%)
Male, n (%)	17 (29.8%)	11 (42.3%)	6 (19.4%)
Foreign background, n (%)	5 (8.8%)	4 (15.4%)	1 (3.2%)
Accelerometer data			
Wear time days, mean (SD)	4.9 (1.7)	4.9 (1.6)	5.0 (1.8)
Wear time hours/day, mean (SD)	13.6 (1.3)	14.0 (1.1)	13.3 (1.5)
LPA min/day, mean (SD)	121.8 (30.7)	134.3 (28.5)	111.3 (28.8)
MVPA min/day, mean (SD)	57.4 (19.3)	62.2 (20.0)	53.4 (18.1)
ST min/day, mean (SD)	638.0 (78.3)	642.6 (68.1)	634.1 (86.9)
Steps/day, mean (SD)	7181 (2093)	8006 (2166)	6490 (1784)
>60 min daily MVPA, n (%)	23 (40.3%)	11 (42.3%)	12 (38.7%)
Commuting to school			
Active commuting, n (%)	6 (10.5%)	3 (11.5%)	3 (9.7%)
Mixed commuting, n (%)	22 (38.6%)	11 (42.3%)	11 (35.5%)
Inactive commuting, n (%)	29 (50.9%)	12 (46.2%)	17 (54.8%)
Km to school, mean (min, max)	9.8 (1.0, 50)	8.6 (1.0, 45)	12.6 (1.6, 50)
Organized sports, n (%)	32 (56.1%)	16 (61.5%)	16 (51.6%)

Abbreviations; LPA = light physical activity; MVPA = moderate to vigorous physical activity; ST = sedentary time; km = kilometers.

**Table 2 ijerph-18-11041-t002:** Within-group changes in physical activity and sedentary time from baseline to 12-month follow-up.

	Intervention				Control			
	N	Mean (95% CI)	Change from Baseline (95% CI)	*p*	N	Mean (95% CI)	Change from Baseline (95% CI)	*p*
MVPA								
Baseline	31	54.7 (48.2, 61.3)	-	-	26	58.6 (51.9, 65.4)	-	-
6 months	31	57.8 (49.5, 66.1)	3.1 (−3.0, 9.2)	0.323	26	55.8 (47.5, 64.1)	−2.8 (−9.2, 3.6)	0.391
12 months	31	58.1 (50.4, 65.8)	3.3 (−3.9, 10,6)	0.367	26	52.1 (45,5 −58,7)	−6.5 (−14.3, 1.3)	0.102
LPA								
Baseline	28	111.8 (101.9–121.7)	-	-	25	130.4 (118.7, 142.0)	-	-
6 months	28	105.2 (93.6–116.8)	−6.6 (−18.6, 5.5)	0.285	25	117.8 (105.2, 130.4)	−12.6 (−23.7, −1.4)	0.027
12 months	28	118.6 (105.2–132.0)	6.8 (−4.2, 17.8)	0.228	25	120.4 (111.7, 129.1)	−9.9 (−19.6, −0.2)	0.044
ST								
Baseline	26	631.8 (617.5, 646.1)	-	-	25	609.5 (594.6, 624.3)	-	-
6 months	26	635.2 (619.4, 651.1)	3.5 (−10.1, 17.1)	0.618	25	624.8 (608.4, 641.2)	15.4 (0.9, 29.8)	0.037
12 months	26	621.7 (602.5, 640.9)	−10.1 (−25.6, 5.4)	0.203	25	625.8 (613.6, 637.9)	16.3 (1.8, 30.8)	0.028

Abbreviations; MVPA = moderate to vigorous physical activity; LPA = light physical activity; ST = sedentary time.

## Data Availability

Data is available upon reasonable request provided approval from appropriate authorities.

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
