# Peer review of "Physical Activity through a Classroom-Based Intervention: A Pragmatic Non-Randomized Trial among Swedish Adolescents in an Upper Secondary School"

_ijerph, 2021, doi:10.3390/ijerph182111041_

Round 1
Reviewer 1 Report
This paper aimed to increase accelerometer- measured PA in a group of students aged 16-18 years with following a 12-month secondary school- based intervention.
While I commend the authors for conducting this study, I consider that there are several points where the study should improve.
The comments for study follow:
The paper I think that has a medium originality because there are many studies carried out on physical activity in adolescents. The strengths that stand out compared to other articles are the ages of the participants (higher) and the use of accelerometers as an objective method to measure Physical Activity for 12 months.
Abstract:
-Statistical analysis of the data is missing.
-On the other hand, it is necessary to unify the ages throughout the study. 16-17 on lines 15 and 57, 16-18 on lines 49 and 171...
Introduction:
-I think you have little information. I should add data to make it more soundness
Materials and Methods:
In Study Desing and Participants:
-Did the additional 2-3 classes of PE have the same characteristics as the usual ones? Were those classes held outside the school center?
Instrumentation and Procedure:
-An image representing the procedure would make it easier to understand.
-Was any type of difference found between the three subgroups based on the material used (standing tables, desktop bikes and regular tables)? It´s a pity that in the limitations the authors have explained that the participants did not always use the same material in those 6 weeks.
-The other cited material, was used in these breaks?
Results:
-Tables and figure are missing. (Table 1. Line 133) (Table 2. Lines 162 and line 168).
-Figure 1 (line 151) is missing.
-Was the impact of PA on the mental health of the participants assessed with reflection time in PE classes?
Discussion:
-I think you have little information. I should add data to make it more soundness.
Strenghts and Limitations:
-They comment that they included measures through seasonal variations, where are they?
Reviewer 2 Report
Abstract
The abstract is well delineated, starting with the objective of the study, the theme is very important, since more and more people practice less sport and at the same time, in these adolescent developmental ages, it is even more important. The sample has an acceptable size, the authors took care to make an experimental group and a control group. The statistics used are adequate. The most important results and conclusions are presented, even if there are no significant statistical differences, this study can be seen as a reminder of this issue. The lack of physical activity at these ages and the time that students spend in a sedentary lifestyle, not respecting the World Health Organization.
Introduction
The authors present the state of the art, from general to specific, related to this theme, raising some problems that persist today. However, this could be a little more elaborate, with the inclusion of some justifications or studies carried out or precisely the lack of them.
Methods
The methodology is well described, where the authors present a considerable sample, the division of the control group and the experimental group. The division of groups was made with different numbers, but close. The study design, procedures, instruments used, and approval of the study by the ethics committee are also presented. Active and Participating by Innovative Classrooms is then carried out with application in a small town in the middle of Sweden, as mentioned by the authors.
The description of the instruments used, the use of a questionnaire to collect anthropometric data and anthropometric data of the participants was also presented.
There was the concern of the authors to explain the place where the instruments were placed. Knowing that at these ages, any foreign element to the body implies some concerns and if this care is not taken, the results can be conditioned. The statistical procedures used were also presented.
Results
Regarding the results, the authors presented different readings of the data obtained: Baseline Characteristics, Between-group Differences in PA, and Within-group Changes in PA. Where it can be observed that in terms of participation in both groups in relation to gender, there is a higher percentage of women in the experimental group. This can even condition or possibly give different results. However, this was not a study objective.
In the end, there are no significant differences at the various moments of analysis, except in both secondary outcomes at baseline in Between-group Differences in PA.
Discussion
Moving on to the discussion, and after analyzing the data achieved during the 12 months. No significant differences were found at the various times studied and compared. The positive but not statistically significant between-group effects on PA in the present study is in line with several previous school-based PA interventions. On the other hand, the authors report that there is a large proportion of previous school interventions reporting positive effects on PA are based on subjective measures.
Finally, comparisons with results from studies with subjective measures are challenging as the findings may differ due to reliability and validity. As well as an additional challenge is that relatively few previous interventions have targeted older adolescents specifically, since the effectiveness of interventions may vary between age groups.
Strengths and Limitations
The agreement with the authors, a major strength of this pragmatic trial was the use of accelerometers to objectively measure PA, with greater reliability and validity as compared to self-report methods. However, accelerometers have a limited ability to capture movements such as swimming, weight training, or bicycling, which may result in an underestimation of PA.
In the swot analysis carried out by the authors, there are positive aspects and others that, despite not being perfect, served to stimulate, evaluate and see what could be improved and changed. It seems to be an important aspect to consider. From the sample number, the percentage balance in relation to gender seems to be another important aspect to be removed, as well as the increase in time and volume of physical activity. Because if there is no greater stimulus, there is no adaptation. As with training and exercise prescription. If one of the main problems of teenagers these days are sedentary lifestyle and lack of physical activity. Since the school is a place of choice for a possible increase in physical activity, then there is something really to change, it is essential.
Conclusions
Finally, regarding the conclusions, although no significant differences were found, it is concluded that this relation within-group changes may suggest that the intervention had a preventive impact on the transition from PA to ST that typically occurs during adolescence.
Bibliography
The bibliography is in accordance with the norms, they are current and they are within the studied theme.
The study presented has a very important relevance with regard to the topic presented. The instruments can be limiting for some of the activities such as swimming, .., despite having been carried out for 12 months (knowing that at these ages, students/teenagers are not always able to use these instruments every day, because it implies an element external to the body).
The study is very interesting and served to quantify what needs to be done and what needs to be changed.
The introduction can and should be changed, improved, with more studies, to be more complete.
The methodology can be a little more complete with the presentation of the intervention program (graphic study design/drawing, as were the exercises, number of repetitions, etc). Giving readers the opportunity to more effectively visualize how the program worked.
The authors spoke about the limitations and strengths of the study. However, it was important to add the practical implications at the end of the study for the students, what they can do and what can be improved and some guidelines to be carried out in schools, in Physical Education classes and in relation to the daily lives of adolescents.
Round 2
Reviewer 1 Report
I think that the authors have been substantial changes made in the article that it has improved.
Just I add that figure 1 would be a table and not figure. I think that would be better a figure but you could leave the table.